# Combined Simplified Molecular Classification of Gastric Adenocarcinoma, Enhanced by Lymph Node Status: An Integrative Approach

**DOI:** 10.3390/cancers13153722

**Published:** 2021-07-24

**Authors:** Till Daun, Ronny Nienhold, Aino Paasinen-Sohns, Angela Frank, Melanie Sachs, Inti Zlobec, Gieri Cathomas

**Affiliations:** 1Institute of Pathology, Cantonal Hospital Basel-Land, 4410 Liestal, Switzerland; till.daun@ksbl.ch (T.D.); ronny.nienhold@ksbl.ch (R.N.); aino.paasinen@unibas.ch (A.P.-S.); angela.frank@ksbl.ch (A.F.); melanie.sachs@ksbl.ch (M.S.); 2Institute of Pathology, University of Bern, 3008 Bern, Switzerland; inti.zlobec@pathology.unibe.ch

**Keywords:** gastric cancer, prognosis, molecular classification, immunohistochemistry, in situ hybridization, next generation sequencing, lymph node metastasis

## Abstract

**Simple Summary:**

In this study, we present a simple but comprehensive molecular analysis of gastric carcinoma. The two major existing classification schemes show some discrepancies and are highly technically demanding, which makes them hardly feasible in daily diagnostic routines. Our workflow is based on simple and commercially available technology and provides a potential consensus approach by integrating the two major classification schemes. Furthermore, our approach allows the molecular subtypes to be assigned to different prognostic groups. We are convinced that our approach may help to better understand the molecular mechanisms of this worldwide health burden and that it could pave the way for new therapeutic targets.

**Abstract:**

Gastric adenocarcinoma (GAC) is a heterogeneous disease and at least two major studies have recently provided a molecular classification for this tumor: The Cancer Genome Atlas (TCGA) and the Asian Cancer Research Group (ARCG). Both classifications quote four molecular subtypes, but these subtypes only partially overlap. In addition, the classifications are based on complex and cost-intensive technologies, which are hardly feasible for everyday practice. Therefore, simplified approaches using immunohistochemistry (IHC), in situ hybridization (ISH) as well as commercially available next generation sequencing (NGS) have been considered for routine use. In the present study, we screened 115 GAC by IHC for p53, MutL Homolog 1 (MLH1) and E-cadherin and performed ISH for Epstein–Barr virus (EBV). In addition, sequencing by NGS for TP53 and tumor associated genes was performed. With this approach, we were able to define five subtypes of GAC: (1) Microsatellite Instable (MSI), (2) EBV-associated, (3) Epithelial Mesenchymal Transition (EMT)-like, (4) p53 aberrant tumors surrogating for chromosomal instability and (5) p53 proficient tumors surrogating for genomics stable cancers. Furthermore, by considering lymph node metastasis in the p53 aberrant GAC, a better prognostic stratification was achieved which finally allowed us to separate the GAC highly significant in a group with poor and good-to-intermediate prognosis, respectively. Our data show that molecular classification of GAC can be achieved by using commercially available assays including IHC, ISH and NGS. Furthermore, we present an integrative workflow, which has the potential to overcome the uncertainty resulting from discrepancies from existing classification schemes.

## 1. Introduction

Gastric cancer is the fifth most common cancer worldwide and the third most common cause of cancer-related death, with over one million new cases annually and more than three-quarters of patients succumbing to the disease [1]. The prognosis remains poor, despite implementation of standardized surgical procedures, reduced postoperative mortality and increased use of multimodal treatment, reaching a 5-year overall survival rate of 20–25% and a median survival of approximately 24 months [2,3,4,5,6].

Over 95% of gastric cancers are adenocarcinomas (GAC) [7,8]. Classically, GAC is classified histologically most widely by the Lauren classification, introduced in 1965 [9] and recently updated by further clarifying the histological features [10]. Basically, two categories are discerned: the intestinal type, associated with chronic atrophic gastritis, followed by intestinal metaplasia and progressive dysplasia, often induced by chronic Helicobacter infection, and the diffuse type with less obvious precursor lesions and inferior clinical outcomes. In addition, the WHO proposed a further histological classification, based on various morphological patterns but without prognostic impact [7].

As in many other tumors, molecular analyses have revealed a new insight into the pathogenesis of GAC, primarily in the search for predictive markers to guide targeted therapy [8]. In GAC, HER-2/neu, a member of the epidermal growth factor family and overexpressed or amplified in various other tumors including breast, colon and other adenocarcinomas, has been shown to be a useful marker for anti-HER-2 antibody therapy [11]. In addition, mismatch deficient tumors characterized by microsatellite instability (MSI), either sporadic or in the context of Lynch syndrome, have been successfully treated with checkpoint inhibitors [12,13]. More recently, molecular classifications for GAC have been proposed, incorporating a comprehensive and broad range of technologies. In 2014, using six different analytic platforms, the Cancer Genomic Atlas (TCGA) research network project presented four different subtypes of GAC, including Epstein–Barr virus (EBV)-associated, MSI, genomically stable (GS) and chromosomally instable (CIN) tumors [14]. Shortly after, the Asian Cancer Research Group (ACRG) presented a further classification, again with four subgroups, namely MSI-high, microsatellite stable/epithelial mesenchymal transition (MSS/EMT), MSS/p53 intact and MSS/p53 loss [15]. Therefore, the two classifications show some overlap but also some mismatches and only the ACRG classification primarily provides prognostic information, although subsequent analysis also showed prognostic significance for the TCGA classification [16]. However, the technology applied for these classifications of GAC is highly technically demanding, costly and time consuming. Therefore, alternative approaches have been proposed, mainly by using immunohistochemistry (IHC) and in situ hybridization (ISH) [17,18,19,20,21]. EBV-associated and MSI tumors can reliably be evaluated by using ISH and IHC, respectively. For p53, however, the exclusive use of IHC has proven to be inadequate and additional approaches have been requested [22,23].

In this study, we present a combined approach using ISH, IHC and commercially available molecular tests to classify GAC based on the TCGA and ACGR classification including the lymph node status and propose a simplified but integrated molecular classification of GAC, including prognostic subgroups.

## 2. Material and Methods

### 2.1. Study Cohort

The study cohort includes 115 patients with primary gastric adenocarcinoma, resected at the Cantonal Hospital Baselland, Switzerland, between May 2002 and October 2016, from whom sufficient tumor blocks were available. Material was limited in 17 of 132 (12.9%) initial cases who were, therefore, excluded from the study. At the time of the evaluation, the tumor blocks were between a few weeks and 14 years old. They had been stored continuously in our archive at ambient temperature and humidity under light protection. Clinical and pathological data were retrieved from the archives of the Institute of Pathology and the Department of Surgery and Oncology of the Cantonal Hospital Baselland, Switzerland. For tumor staging, the 8th edition of the TNM classification of malignant tumors was applied. The study was carried out in accordance with the guidelines of the Cantonal Ethics Committee Basel (Project ID 2018-01065).

### 2.2. Preparation of a Next-Generation Tissue Microarray (ngTMA^®^)

All H&E slides of each case were reviewed and the most representative corresponding tissue blocks (one or two per case) were retrieved from the archives of the Institute of Pathology. Each block was sectioned at 3 µm using standard protocols, scanned using a digital slide scanner (P250 Flash III, 3DHistech, Budapest, Hungary) and uploaded onto a web-based scan management system (Case Center, 3DHistech, Budapest, Hungary). Each scan was annotated, using a tissue microarray (TMA) annotation tool to sample 0.6 mm cylinders. Donor blocks were loaded into the automated tissue microarray (TMA Grandmaster, 3DHistech, Budapest, Hungary), and an image was taken. These images were aligned to the annotated donor block using the accompanying software. A ngTMA was constructed by coring out each annotated region from the donor block and transferring to the recipient block. To avoid bias due to sampling, up to 6 punches were taken from each tumor (4 punches from the center of the tumor, 2 from the invasion front), depending on the available amount of tumor tissue. [24]

### 2.3. Performance and Scoring of Immunohistochemistry (IHC) and In Situ Hybridization (ISH) 

We performed immunohistochemical (IHC) staining for MLH1 (Clone G168-15, BD Biosciences, San Jose, CA, USA), E-Cadherin (E-cad; Clone NCH-38, Agilent Technologies, Santa Clara, CA, USA) and p53 (Clone DO-7, Agilent Technologies, Santa Clara, CA, USA) of the TMA slides. Each punch of the TMA was independently evaluated twice by two experienced pathologists. In cases of inconsistent results between the individual punches of a tumor or complete lack of tumor tissue, immunohistochemistry was repeated and validated in whole slide sections. Loss of MLH1 and p53 expression required a lack of expression in all tumor cells and retaining of wild type expression levels in adjacent normal cells, such as enterocytes, stroma cells or lymphocytes. Strong nuclear expression of p53 in at least 80% of tumor cells was recorded as p53 overexpression. Aberrant expression of E-cad was considered in the case of complete loss or marked reduced membrane staining in >30% of tumor cells [20].

The presence of EBV was tested by the detection of EBV-encoded small RNA 1 (EBER1), using a commercially available in situ hybridization assay (Leica EBER, catalogue number: PB0589, BioSystems, Nunningen, Switzerland). Hybridization was performed in an automated stainer (Bond-Max Leica, BioSystems, Nunningen, Switzerland), using the Bond III Protocol according to the manufacturer’s instructions. Hybridization was visualized by Polymer Retine Red Detection (Cat. No. DS9390, BioSystems, Nunningen, Switzerland) and all slides were counterstained with Haematoxylin. TMAs were analyzed by bright field microscopy and only a strong, positive red nuclear signal was considered positive (Figure 1).

### 2.4. Next Generation Sequencing (NGS) Analysis

For NGS analysis, paraffin was removed from unstained slides by serial treatment with 100% xylol, 100% and 96% ethanol. The tumor cells were enriched from surrounding normal tissue by manual microdissection. For DNA extraction, scraped off tumor cells were dissolved in 50 µL lysis buffer (2.5 µL 1 M TRIS HCl pH 8.5, 0.1 µL 0.5 M EDTA pH 8.0, 0.25 µL Tween 20). A total of 15 µL proteinase K (Cat. No. 19133, Qiagen, Hilden, Germany) was added to the lysate, followed by an incubation for 1 h at 56 °C, 1 h at 90 °C and 5 min at 95 °C. Qubit dsDNA HS Assay Kit (Cat. No. Q32854, Thermo Fisher Scientific Inc., Waltham, MA, USA) was used for the quantification of DNA.

In total, 10 ng of extracted DNA was analyzed by next-generation sequencing (NGS) using the Oncomine Focus Assay (Cat. No. A29230, Thermo Fisher Scientific Inc., Waltham, MA, USA) and the Ion AmpliSeq Colon and Lung Cancer Research Panel v2 (Cat. No 48744413, Thermo Fisher Scientific Inc., Waltham, MA, USA), enabling the detection of somatic mutations and somatic copy number alterations (SCNA) of known cancer hotspot genes (Appendix A). In brief, amplicon libraries were prepared using AmpliSeq Library 2.0 (Cat. No. 4475345, Thermo Fisher Scientific Inc., Waltham, MA, USA) and Ion Xpress Barcode Adapters (Cat. No. 4471250, Thermo Fisher Scientific Inc., Waltham, MA, USA). The samples were equimolarly pooled to reach 550′000 reads per DNA library. The chip loading and emulsion PCR was automated by Ion Chef Instrument (Cat. No. 4484177, Thermo Fisher Scientific Inc., Waltham, MA, USA), sequenced on Ion S5XL System (Cat. No. A27214, Thermo Fisher Scientific Inc., Waltham, MA, USA) and analyzed using the Ion Reporter Software 5.2 (Thermo Fisher Scientific Inc., Waltham, MA, USA). SCNAs were detected by 5% CI value of >4 for amplification and 95%CI value of <1 for deletion.

### 2.5. Statistical Analysis and Presentation of Data

Survival curves were generated using the Kaplan–Meier univariate method and compared by the Mantel–Cox log-rank test. Group comparisons were performed using one-way analysis of variance (ANOVA) or chi-square test. All statistical analyses were calculated within GraphPad Prism v6 and *p*-values of *p* < 0.05 were considered as statistically significant. Graphs were generated in GraphPad Prism v6, while the Oncoprint panel and tables were generated in Microsoft Excel 2016 and supported by conditional formatting. Multiple graphs were arranged using Adobe Illustrator CS6.

## 3. Results

### 3.1. Patient Data, Histology, In Situ Hybridization and Immunohistochemistry

Tumors of a total of 115 patients who underwent surgery for gastric cancer were included in the study. Clinical and pathological characteristics of the patients and tumors at the time of resection are summarized in Table 1. GAC were more common in males (64.3%) and most of the tumors were located in the antrum (47.0%). Intestinal/tubular histology was observed most commonly (72.2%/69.6%) and in 72 patients (62.6%), lymph node metastases were present.

In 2 (1.7%) of 115 tumors, an EBV association was detected by a strong positive EBER ISH. A total of 20 (17.4%) of 115 cancers showed a loss of MLH1 expression in the tumor cells and, therefore, were scored as MSI. Aberrant expression of E-cad was observed in 10 (8.7%) of the 115 tumors, with all of these tumors showing a poorly differentiated histology, either exclusively (*n* = 3) or mixed with areas of poor differentiation (*n* = 7). Aberrant expression of p53 was observed in 52 (45.2%) of the tumors, including 40 (34.7%) tumors with overexpression and 12 (10.4%) with a complete loss of p53 immunostaining (for further details see below). All tumors with aberrant E-cad expression were EBV-negative, but in two tumors, concurrent MLH1 loss was observed (Appendix A). Representative ISH and IHC stains are shown in Figure 1.

### 3.2. Molecular Analysis by Next Generation Sequencing (NGS)

Molecular alterations were analyzed in all tumors by NGS using panels covering the most common altered genes in solid tumors including *TP53*. *TP53* was the most frequently mutated gene, with 63 *TP53* mutations detected in 61 (53.0%) of the total 115 GAC, most commonly seen between amino acid position 130 to 310 (Figure 2a). Most of these 63 *TP53* mutations are classified as missense (44 of 63, 72.1%), followed by nonsense (10 of 63, 16.4%) and frameshift mutations (7 of 63, 11.5%). In three cases, an in-frame deletion of three base pairs was detected, which results in the removal of a single amino acid. Further analysis of the NGS data revealed that other frequently mutated genes are members of the receptor tyrosine kinase pathway, including *PIK3CA* (14 of 115, 12.2%), *KRAS* (13 of 115, 11.3%), *BRAF* (3 of 115, 2.6%) and *AKT1, MAP2K1* and *MTOR* (1 of 115, 0.9%).

In addition to point mutations and small indels, somatic copy number alterations (SCNA) were also detected by the workflow applied (Appendix A). Most frequently, SCNAs were observed in KRAS (13 of 115, 11.3%), MYC (6 of 115, 5.2%) and CCND1 (4 of 115, 3.5%).

### 3.3. Aberrant p53 Expression and TP53 Genetic Alterations

Comparing the results of p53 IHC and the *TP53* genetic analysis, not all patients with loss or overexpression of p53 by IHC carried mutations in the *TP53* gene. In our cohort, 7 (58.3%) of 12 cases with a loss of p53 expression detected by IHC also carried a mutation in *TP53* (Figure 2b). All of the mutations detected in these cases were classified as nonsense or frameshift mutations, resulting in a truncation of the p53 protein. In 5 (41.7%) of 12 cases with loss of p53 expression we did not detect any *TP53* mutation. The majority of patients (37 of 40, 92.5%) with a p53 overexpression showed a missense mutation in *TP53*. Nonsense and frameshift mutations were not detected in patients with p53 overexpression. Only 3 (7.5%) of 40 patients with p53 overexpression did not carry any mutation in the targeted region of *TP53*. By contrast, in 17 (27.0%) of 63 patients with a wild type p53 expression, mutations in *TP53* were detected including frameshift (*n* = 3), nonsense (*n* = 7) and missense mutations (*n* = 7; Figure 2b). Taken together, a total of 76 (56%) of the tumors showed a p53 aberration by either IHC or molecular analysis and were considered TP53/p53 (subsequently named P53) aberrant in the subsequent analysis.

### 3.4. Classification of GAC Subtype, Based on the TCGA and ACRG Classifications 

Based on the two major classification schemes of GAC by TCGA and ACRG as well as further literature [17,18,19,20,21], we classified the GAC of this study in the following algorithm (Figure 3 and Figure 4): MSI and EBV-positive tumors were classified according to the immunohistochemical loss of MLH1 and positive ISH for EBER, respectively. Tumors with loss of E-cad expression by IHC were considered equivalent to the MSS/EMT subtype and GAC with intact p53 and full expression of E-cad surrogated for GS and MSS/p53^+^ tumors. Finally, tumors with aberrant P53 were grouped in the CIN and MSS/p53^−^. In addition, for prognostic purposes, we split the CIN/pp53^−^ in CIN^low^ and CIN^high^ (see below and Table 2 and Figure 2). In the EBV-associated tumors, which include only two patients, the characteristic PIK3CA mutation was present. The MSI subtype, which is identical in both major classification systems, showed a significantly higher mutation rate (excluding the subtype defining *TP53* mutations) compared to the microsatellite stable GACs (2.3 vs. 0.4 per tumor; *p* < 0.001). MSI tumors are linked to an increased age and the EMT subtype to diffuse or mixed histology with poor differentiation (Table 2).

### 3.5. Survival Analysis within the Classifications of GAC Subtypes and Implementing Lymph Node Metastasis as a Prognostic Marker

We further analyzed the survival of this cohort using our modified approach with the two major molecular classification algorithms of TCGA and ACRG (Figure 5a). Whereas the subtypes classified according to the TCGA did not show any difference in prognosis, the ACRG algorithm revealed a significant survival benefit for the MSI compared to the EMT subtype (Figure 5b,c). With 76 (66%) patients, the CIN/p53^−^ subtype was the largest subtype. To reach an improved prognostic stratification, we further subdivided this subtype with respect to the presence of lymph node metastasis, a well-known prognostic factor which has been used in other gene expression tests, for example in breast cancer [25]. Overall, in our cohort, 72 (62.6%) of 115 patients were diagnosed with lymph node metastasis, associated with a significant inferior outcome (Figure 6a). However, no significant survival difference was seen in patients with positive lymph node metastases in MSI tumors and in tumors without p53 alterations (GS/p53^+^; Appendix A). In contrast, in CIN/p53 aberrant tumors, a trend towards an infaust prognosis was observed (*p* = 0.06; Figure 4b). Therefore, we subdivided the CIN/p53^−^ tumors according to the presence of lymph node metastasis, named CIN^low^ without and CIN^high^ with metastases. A significantly better survival for the MSI subtype compared to both the EMT and the CIN^high^/p53^−^ subtypes was observed (Figure 5b). Finally, to receive a better discrimination with respect to prognosis, we grouped the EMT-like and CIN^high^/p53^−^ subtypes into a high risk, and the MSI, EBV and CIN^low^/p53^−^ into a low risk prognostic subgroup with a median survival of 51.0 months and 167.8 months, respectively (*p* = 0.0006; Figure 6b).

## 4. Discussion

In the present study, we classified 115 GAC, using commercially available platforms, including IHC, ISH and NGS, according to the major molecular classification algorithm of TCGA and ACRG. In a second step, we combined the two major classifications, generating five subtypes of GAC: (1) MSI, (2) EBV positive, (3) EMT-like, (4) tumors with aberrant p53 with and without lymph node metastasis and (5) tumors with absence of the major marker listed (Figure 4). Finally, we grouped these subtypes into two prognostic subgroups, one with good-to-intermediate and a second with poor prognosis (Figure 3).

GAC is a heterogeneous tumor and many attempts have been made to classify these tumors for better understanding of the pathogenesis and to improve the treatment and management of the disease [8]. As histology remains the basic step of the primary diagnosis of GAC, histological classifications such as the Lauren or the WHO classification still have value, for example to direct the surgical procedure. More recently, molecular analysis has revealed a more detailed insight into the pathogenesis of GAC, as well as for the targeted treatment of the tumor. The most comprehensive and detailed molecular analyses have been performed by two groups, the TCGA and ACRG, using various platforms, including somatic copy number analysis, whole exome sequencing, DNA methylation profiling, messenger RNA sequencing, micro RNA sequencing, and reverse-phase protein array profiling (TCGA), as well as gene expression profiling, genome-wide copy number microarray and targeted gene sequencing (ACRG) [14,15]. These classifications, however, do have two drawbacks: first, the technical performance for this analysis is highly demanding, complex, time-consuming and costly and has been developed in fresh frozen tissue. Therefore, they are only applicable in a limited manner to the formalin fixed, paraffin embedded tumor tissue usually available in clinical practice. Second, although both classifications define four molecular subtypes of GAC, these subtypes only show a partial overlap, resulting in uncertainty, as a flawless, unifying classification is still lacking [26,27,28].

To overcome the first of these problems, several simplified approaches have been reported by using immunohistochemistry and in situ hybridization [17,18,19,20,21]. Today, detecting EBV by EBER ISH is considered the gold standard for detection of EBV-associated cancer and can easily and reliably be used to diagnose any associated tumors [29]. For MSI tumors, the loss of expression of mismatch repair proteins (MMR) by IHC has been shown to be a sensitive marker, although the latter may miss a small group of patients, namely those with Lynch syndrome [30]. Patients with EBV-associated GAC and MSI tumors show a better prognosis and high rate of response following treatment with immune checkpoint inhibitors, due to the generation of neoantigens and high expression of PD-L1 [12,13,31,32].

More complex is the IHC of p53 for GAC, which has been used as surrogate marker for the CIN subtype in the TCGA, which showed a genetic aberration of TP53 in 71%, and for the MSS/p53^−^ subtype in the ACRG classification [17,18,19,20,21]. Aberrant expression of p53 by immunohistochemistry has been assessed differently: some authors exclusively rated p53 overexpression of tumor cells, using different cutoffs [17,19,21], whereas others included the loss of p53 as aberrant expression in their analysis [18,20]. In an extensive and sophisticated study, Schoop et al. show that p53 IHC alone cannot predict *TP53* mutations for the CIN subtype of GAC’s, and additional molecular analysis has been demanded [22,23]. In our study, we therefore combined p53 IHC, including loss and overexpression, and *TP53* gene sequencing using a commercially available NGS technique applicable to formally fixed, paraffin embedded tissue [33]. We showed that in tumors with aberrant p53 expression, IHC is discriminative for the type of *TP53* mutation: NGS identified *TP53* truncation mutations in 60% of patients with loss of p53 expression, serving as a molecular cause for the lack of the p53 signal and evidence for the loss of p53 function. In 90% of patients with p53 overexpression, missense mutations were detected in *TP53*. A recent study suggests that somatic point mutations in *TP53* cause a dominant-negative effect, resulting in loss of p53 function [34]. However, we identified patients with wild type p53 expression, who carry mutations in *TP53*. In this group, nonsense, missense and frameshift mutations were observed, suggesting the existence of tumors with loss of p53 function in spite of wild type p53 expression pattern. In 10% of the patients with p53 overexpression and 40% of the patients with loss of p53 expression we did not detect any *TP53* mutation. This might be explained by the target regions of the NGS panel, which only cover 80% of the *TP53* coding region. Other reasons for loss of p53 expression with a lack of *TP53* mutations could be chromosomal aberrations, such as focal deletions of the *TP53* gene or loss of chromosome 17q13. These results are in line with recent observations in ovarian cancer, whereas studies in GAC showed a lower prevalence of *TP53* mutations in cases with aberrant p53 expression [35,36,37]. With the combination of IHC and NGS we improved the identification of patients with aberrant *TP53*. 

Is it scientifically sound to simply combine the two classifications? One must keep in mind that a limited number of IHC assays or a restricted NGS analysis represent surrogate markers of the extensive analyses of the TCGA and ACRG platforms. As mentioned above, for MSI and EBV, the MMR IHC and ISH reach a high and very high accuracy, respectively. The MSI subtype is similar in both classification systems, with a rate of 22% and 22%, occurring in 17.4% of our patient cohort. EBV-associated tumors have been assigned a specific subtype only in the TCGA classification with a rate of 9%, but the majority of EBV-associated tumors group in the MSS/p53+ subtype of the ACRG (12/18 EBV positive tumors), and the authors admit that these tumors may represent an own subtype [15]. In our cohort, only two (1.7%) tumors are EBV-associated which is within the worldwide reported range of prevalence and in line with previous data of Switzerland [38,39]. The CIN subtype, with 50% being the most common subtype of TCGA classification, is characterized by chromosomal instability, namely the degree of aneuploidy and a high somatic copy number variation. In addition, 73% of these tumors show a TP53 mutation, bringing the CIN subtype close, but not completely congruent, to the MSS/p53^−^ subtype (35.7%) of the ACRG. In addition, other IHC-based classification systems describe aberrant p53 expression as the major molecular abnormality in GAC, affecting 28% to 51% of patients [17,18,19,20,21].

The loss of E-cad by IHC can be considered a reliable marker for the MSS/EMT subtype of the ACRG, as these GACs are associated with diffuse/mixed histology and associated with a poor outcomes, as has been shown by Ahn et al. and confirmed in our study [20]. In the GS subtype of TCGR, characterized by low mutational burden and somatic copy number aberration, 37% of tumors show mutation in the CDH1 gene, indicating that a subset but not all the GS tumors may overlap with the MSS/EMT [15]. Formally, GS tumor may be included in the MSS/p53^+^ subtype of ACRG; however, in the classification using IHC and ISH, the GS subtype remains a tumor primarily defined by exclusion.

## 5. Conclusions

In summary, our data show that molecular classification of GAC can be achieved by using commercially available assays including IHC, ISH and NGS. Molecular classification is an indispensable prerequisite to better understand the devastating disease of GAC. To overcome the standoff situation of two comprehensive but seemingly concurrent classifications, we propose a combined workflow of TCGR and ACRG, which can be applied by using restricted technologies but hopefully may set a frame for a future comprehensive molecular consensus classification of GAC.

## Figures and Tables

**Figure 1 cancers-13-03722-f001:**
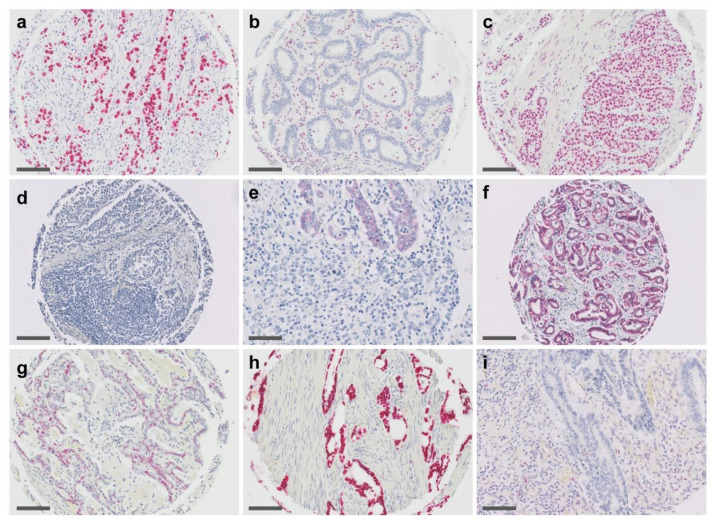
In situ hybridization and immunohistochemistry for gastric adenocarcinoma. (**a**) Epstein–Barr virus (EBV)-encoded RNA (EBER) detected by in situ hybridization in EBV-associated GAC. (**b**) Immunohistochemistry of MLH1 in a microsatellite instable tumor (MSI) presenting with a complete loss of staining in the nuclei of tumor cells and retained expression in adjacent stromal cells. (**c**) Retained MLH1 expression in tumor and stromal and inflammatory cells of a microsatellite stable cancer (MSS). (**d**–**f**) Expression of E-Cadherin: (**d**) Homogenous complete loss for E-Cadherin (note: lymphocytes in lower half of image are endogenously negative for E-Cadherin). (**e**) Heterogeneous expression: areas with glandular differentiation show a weak E-Cadherin expression, whereas the undifferentiated region (lower half of image) shows complete loss. (**f)** Tumor with strong E-Cadherin expression. (**g**–**i**) Expression of p53 in GAC detected by immunohistochemistry: (**g**) Wild type expression showing a mixture of negative cells, weakly as well as strong positive cells. Aberrant expression include (**h**) overexpression with a strong diffuse positivity in ≥80% of tumor cell nuclei and (**i**) complete loss of staining in all tumor cell nuclei with a variable, usually weak positivity in the surrounding stromal cells. Scale bars indicate 100 µm.

**Figure 2 cancers-13-03722-f002:**
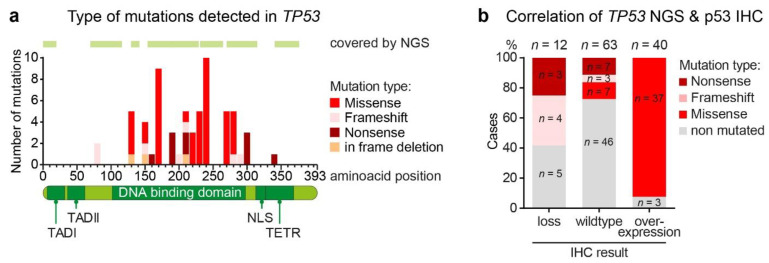
Genetic and immunohistochemical evaluation of *TP53* and p53 in GAC, respectively. (**a**) Distribution of the different types of *TP53* mutation over the TP53 gene. (**b**) Relationship between aberrant p53 overexpression and the various types of *TP53* mutations. TADI, transactivation domain 1; TADII, transactivation domain 2; NLS, nuclear localization signal; TETR, oligomerization domain; IHC, immunohistochemistry; NGS, next-generation sequencing.

**Figure 3 cancers-13-03722-f003:**
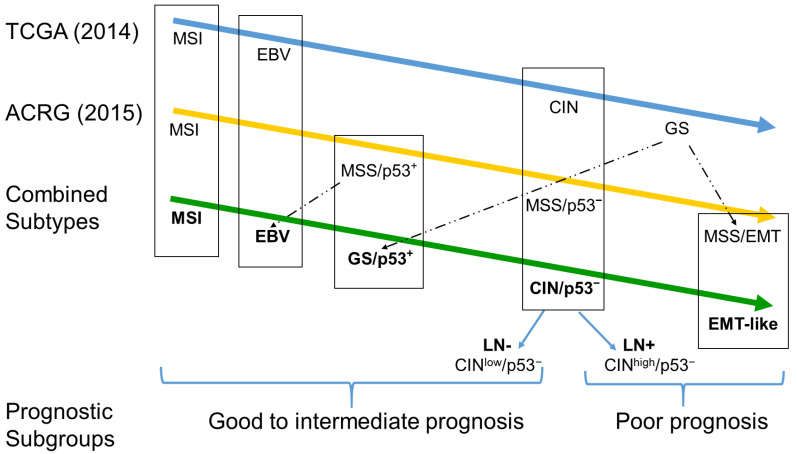
Algorithms of the major molecular classification of GAC, TCGA (blue) and the ACGR (yellow) classification, and the simplified, combined workflow to classify the GCA subtypes used in this study (green). The five subtypes have finally been grouped into two prognostic subgroups with good-to-intermediate and poor prognosis. The nomenclature is based on unique terms (MSI, EBV), similar terms if appropriate (EMT-like) and in a chronologic sequence in case of mixed groups (GS/p53^+^, CIN/p53^−^; TCGA: 2014, ACRG: 2015). LN− and LN+: Lymph nodes without and with metastases, respectively.

**Figure 4 cancers-13-03722-f004:**
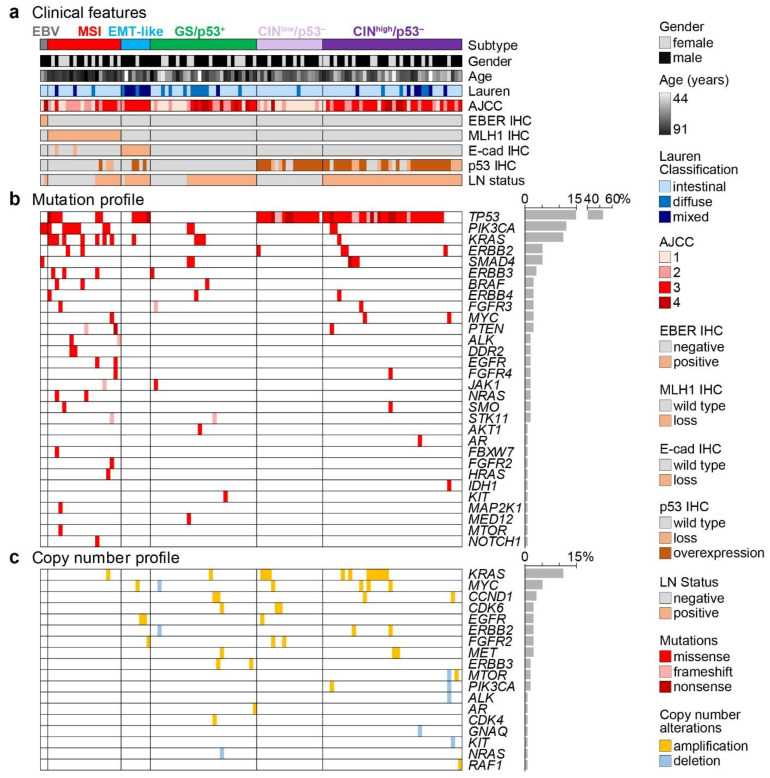
Gastric adenocarcinoma were divided into subtypes: Epstein–Barr virus (EBV)-positive (grey), microsatellite instable (MSI; red), epithelial-mesenchymal transition (EMT)-like (blue), genomic stable/p53 unaffected (GS/p53^+^; green) and chromosomal instable/p53 aberrant (CIN/p53^−^; violet). CIN/p53^−^ tumors were further divided according to the absence (light violet) or presence of lymph node metastasis (dark violet). Oncoprint panel displaying: (**a**) Clinical data, histology and immunohistochemistry, (**b**) Mutation profiles and (**c**) copy number variation.

**Figure 5 cancers-13-03722-f005:**
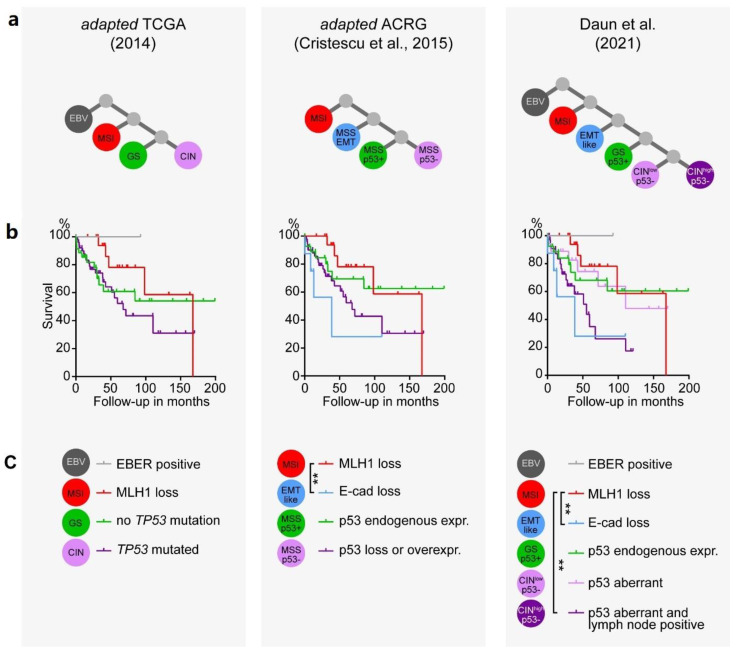
Association of survival and the two major molecular classifications of gastric adenocarcinoma, the TCGA and ACRG and the workflow described in this work, using the data of this study. (**a**) Schematic of the algorithm applied and (**b**) Kaplan–Meier survival curves with (**c**) legend listing subtype-defining markers and statistical significances calculated by pairwise group comparison with the Mantel–Cox log-rank test. Asterisks depict magnitude of statistical significance: **, *p* < 0.01.

**Figure 6 cancers-13-03722-f006:**
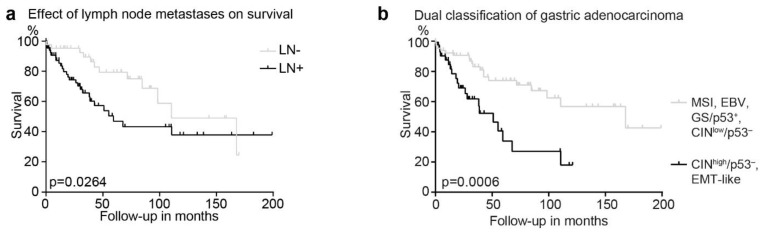
Survival analysis in 115 patients with GAC. (**a**) Impact of the presence of lymph node metastasis in the total cohort shows an inferior outcome in patients with lymph node metastasis. (**b**) Grouping the subtypes of GAC into high and low risk tumors: Low risk: MSI, EBV, and CIN^low^/p53^−^; high risk: CIN^high^/p53^−^ and EMT-like.

**Table 1 cancers-13-03722-t001:** Clinical and histological characteristics of 115 patients diagnosed with gastric adenocarcinoma.

Characteristic	*n*	%
Gender		
male	74	64.3%
female	41	35.7%
Age		
range: median (min–max)	74 (44–91)	
Anatomic region		
GEJ	12	10.4%
Cardia	33	28.7%
Corpus	16	13.9%
Antrum	54	47.0%
Lauren classification		
Intestinal	83	72.2%
Diffuse	17	14.8%
mixed	15	13.0%
WHO classification		
Tubular	80	69.6%
Papillary	1	0.9%
Mucinous	3	2.6%
Poorly cohesive	17	14.8%
Mixed	14	12.2%
Pathologic T	*n*	%
1	21	18.3%
2	24	20.9%
3	37	32.2%
4	33	28.7%
Pathologic N		
0	43	37.4%
1+	72	62.6%
Pathologic N according to AJCC		
0	43	37.4%
1	24	20.8%
2	18	15.7%
3	30	26.1%
Pathologic M		
0	103	89.6%
1	12	10.4%
AJCC stage		
1	32	28%
2	30	26%
3	41	36%
4	12	10%

**Table 2 cancers-13-03722-t002:** Clinicopathological characteristics of molecular subtypes of gastric adenocarcinoma. ^a^ one-way analysis of variance (ANOVA), for all other comparisons Chi-square test was used.

Characteristic	EBV		MSI		EMT-Like		GS/p53^+^		CIN^low^/p53^−^		CIN^high^/p53^−^		*p* Value
Total	n	%	n	%	n	%	n	%	n	%	n	%	
	2	1.7%	20	17.4%	8	7.0%	29	25.2%	18	15.7%	38	33.0%	
Gender													
male	2	100%	10	50.0%	7	87.5%	18	62.1%	13	72.2%	24	63.2%	0.5001
female	0	0.0%	10	50.0%	1	12.5%	11	37.9%	5	27.8%	14	36.8%	
Age													
median (min–max)	82 (61–91)		82 (61–91)		74 (44–81)		74 (44–90)		73 (61–89)		68 (44–84)		0.0004 ^a^
Median survival													
months	undefined		167.8		38.7		undefined		110.5		55.3		0.0372
Anatomic region													
GEJ	0	0.0%	1	5.0%	0	0.0%	2	6.9%	5	27.8%	4	10.5%	0.0288
Cardia	1	50.0%	2	10.0%	3	37.5%	8	27.6%	6	33.3%	13	34.2%	
Corpus	1	50.0%	1	5.0%	0	0.0%	2	6.9%	2	11.1%	10	26.3%	
Antrum	0	0.0%	16	80.0%	5	62.5%	17	58.6%	5	27.8%	11	28.9%	
Lauren classification													
Intestinal	2	100%	17	85.0%	0	0.0%	19	65.5%	17	94.4%	28	73.7%	0.0404
Diffuse	0	0.0%	1	5.0%	2	25.0%	9	31.0%	1	5.6%	4	10.5%	
mixed	0	0.0%	2	10.0%	6	75.0%	1	3.4%	0	0.0%	6	15.8%	
WHO classification													
Tubular	2	100%	17	85.0%	0	0.0%	17	58.6%	16	88.9%	28	73.7%	0.3522
Papillary	0	0.0%	0	0.0%	0	0.0%	1	3.4%	0	0.0%	0	0.0%	
Mucinous	0	0.0%	1	5.0%	1	12.5%	1	3.4%	0	0.0%	0	0.0%	
Poorly cohesive	0	0.0%	0	0.0%	3	37.5%	8	27.6%	1	5.6%	5	13.2%	
Mixed	0	0.0%	2	10.0%	4	50.0%	2	6.9%	1	5.6%	5	13.2%	
Pathologic T													
1	0	0.0%	2	10.0%	0	0.0%	7	24.1%	8	44.4%	4	10.5%	0.1517
2	1	50.0%	4	20.0%	1	12.5%	6	20.7%	6	33.3%	6	15.8%	
3	0	0.0%	8	40.0%	3	37.5%	7	24.1%	4	22.2%	15	39.5%	
4	1	50.0%	6	30.0%	4	50.0%	9	31.0%	0	0.0%	13	34.2%	
Pathologic N													
0	1	50.0%	13	65.0%	1	12.5%	10	34.5%	18	100%	0	0.0%	<0.0001
1+	1	50.0%	7	35.0%	7	87.5%	19	65.5%	0	0.0%	38	100%	
Pathologic N according to AJCC													
0	1	50.0%	13	65.0%	1	12.5%	10	34.5%	18	100%	0	0.0%	<0.0001
1	0	0.0%	2	10.0%	0	0.0%	8	27.6%	0	0.0%	14	36.8%	
2	0	0.0%	3	15.0%	3	37.5%	5	17.2%	0	0.0%	7	18.4%	
3	1	50.0%	2	10.0%	4	50.0%	6	20.7%	0	0.0%	17	44.7%	
Pathologic M													
0	1	50.0%	20	100%	8	100%	23	79.3%	17	94.4%	34	89.5%	0.1113
1	1	50.0%	0	0.0%	0	0.0%	6	20.7%	1	5.6%	4	10.5%	
AJCC stage													
1	1	50%	5	25%	0	0%	9	31%	13	72%	4	11%	<0.0001
2	0	0%	9	45%	1	13%	7	24%	4	22%	9	24%	
3	0	0%	6	30%	7	88%	7	24%	0	0%	21	55%	
4	1	50%	0	0%	0	0%	6	21%	1	6%	4	11%	

## Data Availability

Data are available upon request from the corresponding author.The data presented in this study are available in the manuscript and Appendix A. Any further information is available from the corresponding author.

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
