# Peer review of "Combined Simplified Molecular Classification of Gastric Adenocarcinoma, Enhanced by Lymph Node Status: An Integrative Approach"

_cancers, 2021, doi:10.3390/cancers13153722_

Round 1

Reviewer 1 Report

I have carefully reviewed the manuscript entitled: "Combined simplified molecular classification of gastric adenocarcinoma, enhanced by lymph node status: An integrative approach." by Daun et al. In this work, Daun et al. used commercially available assays, including immunohistochemistry ( IHC), in situ hybridization (ISH) and next-generation sequencing (NGS) to do the molecular classification of Gastric Adenocarcinoma (GAC). I have few concerns that could be addressed to improve the quality of this study.

Major comments:

  1. The manuscript lacks novelty as several simplified approaches have been reported already by using IHC and ISH to do the molecular classification of GIC (e.g., PMID: 27032689, PMID: 29721214, PMID: 31803306).

Minor comments:

  1. Please mention Figure 1 (images of In situ hybridization and immunohistochemistry for GIC images) in section 3.1.
  2. In Figure 2b, please change the color of fonts or the bar plot. Some of the texts are almost invisible.
  3. Tumors with loss of E-cadherin expression by IHC were considered equivalent to the MSS/EMT subtype in this study. Were other EMT markers besides E-cadherin also checked in this subtype of GIC?
  4. In Figure 5, please indicate Figures a, b and c.
  5. In Figure 5, p values of Kaplan-Meier survival curves are missing.
  6. On page 10, line 259, cited article [25] looks irrelevant. Please cite the correct article.

Author Response

Comments of Reviewer 1

I have carefully reviewed the manuscript entitled: "Combined simplified molecular classification of gastric adenocarcinoma, enhanced by lymph node status: An integrative approach." by Daun et al. In this work, Daun et al. used commercially available assays, including immunohistochemistry ( IHC), in situ hybridization (ISH) and next-generation sequencing (NGS) to do the molecular classification of Gastric Adenocarcinoma (GAC). I have few concerns that could be addressed to improve the quality of this study.

Major comments:

  1. The manuscript lacks novelty as several simplified approaches have been reported already by using IHC and ISH to do the molecular classification of GIC (e.g., PMID: 27032689, PMID: 29721214, PMID: 31803306).

We appreciate the comments of the reviewer and would like to respond as follows: First, thanks a lot for the references mentioned: These include one original work (Setia et al, our reference 18) and two very interesting reviews from 2018 and 2019. Choosing references, namely reviews, is somehow a subjective decision, also to keep the number of references in a decent range. In our original manuscript, we included 3 references of reviews that we felt served our purpose, i.e. reference 8 (2017), reference 22 (2018) and reference 26 (2021). We definitively appreciate the paper by Setia and coworkers and paid tribute to their excellent work.

We consider that our study offers new insights by combining commercial molecular TP53 analysis by NGS and IHC for measuring p53 expression to determine aberrant p53 function. Several studies (reference 22 and 23 of our manuscript) as well as our own data presented (Figure 2) have shown that using immunohistochemistry for p53 not really detect all p53 alterations. On the other hand, today, commercial tests made p53 mutation analysis easily accessible and should therefore be included in the algorithm of molecular classification of gastric cancer.

Furthermore, in contrast to other studies in the field, our study applies multiple classification systems (Figure 5) on the same cohort, allowing direct comparison of performance and clinical utility. Finally, our algorithm stratifies the patients in two highly significant prognostic groups and we propose, based on our data, a frame for a consensus classification of gastric adenocarcinomas. Therefore, we are convinced that our data provide a significant number of biological and clinical novelties relevant for the readers of the Journal.

Minor comments:

  1. Please mention Figure 1 (images of In situ hybridization and immunohistochemistry for GIC images) in section 3.1.

We thank the reviewer for pointing out a missing reference to Figure 1 in the main text and added the following sentence at the end of paragraph 3.1: Representative ISH and IHC stains are shown in Figure 1.

  1. In Figure 2b, please change the color of fonts or the bar plot. Some of the texts are almost invisible.

In order to improve the readability of the text within Figure 2b, we changed the font color to black.

  1. Tumors with loss of E-cadherin expression by IHC were considered equivalent to the MSS/EMT subtype in this study. Were other EMT markers besides E-cadherin also checked in this subtype of GIC?

E-cadherin is a well-established marker for EMT (see e.g. reference 18 or 20 in the manuscript). Therefore, we did not use other markers for EMT. We are aware that E-cadherin serves as a surrogate marker and hence used the term of “EMT-like” in our manuscript. Following strict definitions for the evaluation of the E-cadherin immunohistochemistry, we gained extensive experience with this marker and introduced it in the routine diagnostic of gastric adenocarcinomas.

  1. In Figure 5, please indicate Figures a, b and c.

We thank the reviewer for pointing out missing letters to address specific parts in Figure 5 and added A, B and C accordingly.

  1. In Figure 5, p values of Kaplan-Meier survival curves are missing.

In our study, we displayed survival in Kaplan-Meier curves and calculated significant difference in survival by utilizing the Mantel-Cox log-rank test. However, this test only allows to compare two individual groups. In graphs, which only contain two groups, the p-value is displayed in the graph. In graphs with more than two groups, the Mantel-Cox log-rank test for each individual combination of two groups. Due to the lack of space and to enhance the clarity in Figure 5b, p-values are not shown for each pairwise comparison. Instead, groups with a statistical significant survival are connected with a bar and one or more asterisks are added to represent the magnitude of the p-value. We extended the legend of Figure 5 and the paragraph 2.5 (Statistical analysis) and hope to answer this point to the reviewer.

  1. On page 10, line 259, cited article [25] looks irrelevant. Please cite the correct article.

We thank the reviewer for this note and have added the information to the main text that the cited article relates to a gene expression test, which is already used in breast carcinoma.

Reviewer 2 Report

Authors presented molecular analysis of GAC. Although the information is presented clearly, I have a few comments.

  1. Please remove any extra spaces or insert the missing ones.
  2. If the abstract explains the abbreviations, please elaborate on all of them, e.g. MSI
  3. Paragraph 2.1. - It should be clearly written how old are samples, how did you store them.
  4. Did you prepare any control (healthy) group?
  5. How much DNA was used in the research?
  6. Page 4, line 145 -What is the percentage of xylol and ethanol used in research?
  7. Page 5, line 177 - it is not necessary to repeat "Table 1"
  8. Figure 2. - please make the text (N=4, N=5, DNA binding domain etc) a bit more visible. 
  9. Figure 4. - which programme was used for preparing this figure?
  10. Page 10, line 261. - change Fig 6a to Figure 6a.
  11. Figure 5. - please clarify which method did you use for statistical significance
  12. Page 12, line 286 - as I understand you grouped subtypes in three prognostic subgroups, not two?
  13. Did you upload all characteristic and obtained results in any repository?
  14. What were the admission and exclusion conditions for patients? Did they have any comorbidities that might have affected the results? Were they taking any medications?

Despite the questions mentioned above, I believe the this work is worth considering for publication in the journal.

Author Response

Comments of Reviewer 2

Authors presented molecular analysis of GAC. Although the information is presented clearly, I have a few comments.

We are pleased to see that the reviewer 2 consider our manuscript worth considering for publication and we are thankful for the careful reviewing.  The individual points we would like to answer as follows:

  1. Please remove any extra spaces or insert the missing ones.

We apologize for the typos and screened the manuscript for additional grammatical improvements.

  1. If the abstract explains the abbreviations, please elaborate on all of them, e.g. MSI

We thank the reviewer for this note and explained all abbreviations in the abstract.

  1. Paragraph 2.1. - It should be clearly written how old are samples, how did you store them.

We agree to the reviewer with this highly relevant point and have added the age of the samples in the mentioned paragraph and the conditions under which they had been stored.

  1. Did you prepare any control (healthy) group?

For the tests we applied, appropriate positive and negative controls were included.  As the primary goal of the study was to analyze the molecular profiles of gastric cancer and the prognostic impact of these subtypes on survival, we did not include a healthy control group.  We feel that, beside the complexity to define an adequate control group, this control group would not add further information to the primary questions of our work.

  1. How much DNA was used in the research?
  2. Page 4, line 145 -What is the percentage of xylol and ethanol used in research?
  3. Page 5, line 177 - it is not necessary to repeat "Table 1"

Answer to point 5-7: We thank the reviewer for pointing out missing or unnecessary pieces of information. We added the amount of DNA, added relative concentrations of xylol and ethanol and removed the reference to Table 1.

  1. Figure 2. - please make the text (N=4, N=5, DNA binding domain etc) a bit more visible.

In order to improve the readability of the text within Figure 2a and b, we increased the font size, moved domain information to the legend of this figure and changed the font color.

  1. Figure 4. - which programme was used for preparing this figure?

The Figure is a combination of a so-called “oncoprint” panel, displaying clinical data and mutations of individual patients, and a bar graph, showing the frequency of genetic alteration in an individual gene. The oncoprint panel has been generated in Microsoft Excel 2016 (with the help of conditional formatting). The bar graph has been generated in Graphpad Prism v6. Both graphs were combined and a legend was generated by utilizing Adobe Illustrator CS6. We added additional information for the creation of figures in section 2.5 (previously called “Statistical analysis”, now “Statistical analysis and presentation of data”).

  1. Page 10, line 261. - change Fig 6a to Figure 6a.

We apologize for the typos and screened the manuscript for additional grammatical improvements.

  1. Figure 5. - please clarify which method did you use for statistical significance

In our study, we displayed survival in Kaplan-Meier curves and calculated significant difference in survival by utilizing the Mantel-Cox log-rank test. However, this test only allows to compare two individual groups. In graphs, which only contain two groups, the p-value is displayed in the graph. In graphs with more than two groups, the Mantel-Cox log-rank test for each individual combination of two groups. Due to the lack of space and to enhance the clarity in Figure 5b, individual p-values are not shown for each pairwise comparison. Instead, groups with a statistical significant survival are connected with a bar and one or more asterisks are added to represent the magnitude of the p-value. We extended the legend of Figure 5 and the paragraph 2.5 and hope to answer this point to the reviewer.

  1. Page 12, line 286 - as I understand you grouped subtypes in three prognostic subgroups, not two?

We apologize for the imprecise description of the prognostic subgroups. We aimed on defining two subgroups: One with good to intermediate prognosis, which contains the subtypes MSI, EBV, GS/p53+ and CINlow/p53-, and a second subgroup with poor prognosis, containing the subtypes CINhigh/p53-. To improve readability, we improved the legend of Figure 3 and the main text of the original manuscript on line 286. We also updated the figure and moved the label of the prognostic subgroups to the level of the names of the subgroups.

  1. Did you upload all characteristic and obtained results in any repository?

Because the ethical allowance for this study has been granted retrospectively, many of the included patients have passed away and were unable to explain their will for publication of genetic information. Swiss law therefore does not allow upload genetic raw data to public databases. However, genetic data is open upon request. For all other results, we generated a new table (Supplemental table 3), which contains all summarized clinical, pathological and genetic data.

  1. What were the admission and exclusion conditions for patients? Did they have any comorbidities that might have affected the results? Were they taking any medications?

We thank the reviewer for this relevant comment. We included patients from May 2002 to October 2016, who received surgical treatment for removal of primary gastric cancer and from whom sufficient material was available for histologic and molecular analysis. Material was limited in 17 of 132 (12.9%) initial cases and therefore were excluded from the study. This information was added to the manuscript in the material and methods section.

As our study is a retrospective study, comorbidities and medications were not systematically registered and collected and therefore not included in the study. However, in all patients, the decision for surgical treatment of the gastric cancer was made in the multidisciplinary tumor board of our Institution, the Cantonal Hospital Baselland, ensuring an adequate uniformity in the comorbidities of our patients.    

Despite the questions mentioned above, I believe the this work is worth considering for publication in the journal.

Reviewer 3 Report

The manuscript is well structured.

It presents the results of a surgical cohort study.

In order to improve the treatment and management of GAC (Gastric adenocarcinoma), the authors propose an integrative approach to classify this histologically and aetiologically heterogeneous tumor, presenting a simple but comprehensive molecular analysis of gastric carcinoma.

The topic is interesting. It is an important research area. There is currently little evidence in literature and there are still many unanswered questions.

In my opinion, the article can be published.

There are some small typos to fix. A careful reading is suggested. The authors should check the correspondence of the results reported in the text with the ones in the tables (e.g. the percentages of observed intestinal/tubular histology, that are reported in the results section “77.4%/74.8%” and in Table 1 “72.2%/69.6%”).

Author Response

Comments of Reviewer 3

The manuscript is well structured.

It presents the results of a surgical cohort study.

In order to improve the treatment and management of GAC (Gastric adenocarcinoma), the authors propose an integrative approach to classify this histologically and aetiologically heterogeneous tumor, presenting a simple but comprehensive molecular analysis of gastric carcinoma.

The topic is interesting. It is an important research area. There is currently little evidence in literature and there are still many unanswered questions.

In my opinion, the article can be published.

There are some small typos to fix. A careful reading is suggested. The authors should check the correspondence of the results reported in the text with the ones in the tables (e.g. the percentages of observed intestinal/tubular histology, that are reported in the results section “77.4%/74.8%” and in Table 1 “72.2%/69.6%”).

We are very pleased about the positive reviewer comment. 

We apologize for the typos and screened the manuscript for additional grammatical improvements. Indeed, the percentages mentioned in the text represent outdated values, while the table contain correct results.

Round 2

Reviewer 1 Report

The manuscript entitled "Combined simplified molecular classification of gastric adenocarcinoma, enhanced by lymph node status: An integrative approach" described the use of commercially available assays, including immunohistochemistry ( IHC), in situ hybridization (ISH), and next-generation sequencing (NGS), to do the molecular classification of Gastric Adenocarcinoma (GAC). The revised manuscript is well written, and the authors have well addressed my concerns.